# Social prescribing for people living with dementia (PLWD) and their carers: what works, for whom, under what circumstances and why – protocol for a complex intervention systematic review

Jessica Marshall [1], Evie Papavasiliou [2], Chris Fox [3], Matthew Hawkes [4], Anne Irvine,[4] Esme Moniz-Cook [5], Aimee Pick [6], Marie J Polley [7], Joanne Reeve [8], Louise Robinson [9], George Rook,[1] Euan Sadler [10], Emma Wolverson [11], Sarah Walker [3], Jane L Cross [4]

For numbered affiliations see end of article.

**Correspondence to**
Dr Jane L Cross;
j.cross@uea.ac.uk

## ABSTRACT

**Introduction** Dementia is a complex medical condition that poses significant challenges to healthcare systems and support services. People living with dementia (PLWD) and their carers experience complex needs often exacerbated by social isolation and challenges in accessing support. Social prescribing (SP) seeks to enable PLWD and their carers to access community and voluntary sector resources to support them address such needs. Existing research, however, does not describe what SP interventions are currently in place in dementia care. Little is known about the needs these interventions are designed to address, the reasons that lead PLWD and their carers to participate in them, their effectiveness and the extent to which they could increase positive health outcomes if adopted and how.

**Methods and analysis** A complex intervention systematic review of SP for PLWD and/or their carers will be conducted using an iterative logic model approach. Six electronic (MEDLINE, EMBASE, PsycINFO, CINAHL, Scopus and Cochrane/CENTRAL) and two grey literature databases (EThOS and CORE) were searched for publications between 1 January 2003 and June 2023, supplemented by handsearching of reference lists of included studies. Study selection, data extraction and risk of bias assessment, using Gough's Weight of Evidence Framework, will be independently performed by two reviewers. A narrative approach will be employed to synthesise and report quantitative and qualitative data. Reporting will be informed by the Preferred Reporting Items for Systematic Review and Meta-Analysis Complex Interventions extension statement and checklist.

**Ethics and dissemination** No ethical approval is required due to this systematic review operating only with secondary sources. Findings will be disseminated through peer-reviewed publications, conference presentations and meetings with key stakeholders including healthcare professionals, patient and carer groups, community organisations (eg, the Social Prescribing Network and the Evidence Collaborative at the National Academy for Social Prescribing), policymakers and funding bodies.

| STRENGTHS AND LIMITATIONS OF THIS STUDY |
| --- |
| ⇒ This will be the first protocol using complex intervention systematic review methodology to report current social prescribing practice in dementia care. |
| ⇒ Logic modelling will be used to visualise the complexity of social prescribing for people living with dementia and their carers. |
| ⇒ The quality and relevance of findings will be strengthened by the contributions of the SPLENDID Work Package 1 Expert and the Patient & Public Involvement Advisory Groups involved in this study. |
| ⇒ This review may be confined by empirical evidence being scarce or heterogeneous. |
| ⇒ Included studies were limited to those carried out within the UK and reported in the English Language due to the variation in coverage and delivery of social prescribing. |

**PROSPERO registration number** CRD42023428625.

## INTRODUCTION

Dementia is a complex medical condition that poses significant challenges to healthcare systems and support services, affecting millions of people worldwide. In the UK, there are an estimated 944 000 people living with dementia (PLWD),[1] predicted to rise to over 1.1 million by 2030.[2] PLWD and their carers experience multifaceted, complex needs (medical, emotional, social, financial and practical) often exacerbated by social isolation and challenges in accessing support. The inability to consistently meet these needs in a timely manner can result in poorer health outcomes and quality of life.[3] While a dementia diagnosis offered as early as possible is thought to mobilise support for

PLWD, access to post-diagnostic care has been a challenge for people with dementia and their families.[4] This leads to high levels of stress, and distress when stress becomes unresolved.[5]

Wider conceptualisation of health evidence in primary care and public health has advocated for care built on an understanding of wider social determinants of health.[6] One recent articulation is the concept of social health that focuses on three key dimensions (personal, disease-related and social and physical environment influencing factors),[7 8] based on which, social prescribing interventions to counteract social isolation and disengagement for PLWD are built.[9] Social prescribing seeks to enable people with unmet complex needs to access community and voluntary sector resources to support them address their needs using non-biomedical approaches[10] which, in association with optimising social health, could improve the quality of daily living for PLWD and their carers. The diversity of social prescribing interventions offered (psychosocial, psychological, arts-based and educational), the variability in outcome measures assessed and the wide range of assessment instruments used indicate the complex nature of this non-pharmacological approach in dementia care.[7]

Until recently, there was no standard definition of social prescribing, however, a Delphi study in 2022[11] proposes both conceptual and operational definitions. For the purposes of this review, the conceptual definition will be adopted where social prescribing is defined as 'a means for trusted individuals in clinical and community settings to identify that a person has non-medical, health-related social needs and to subsequently connect them to non-clinical supports and services within the community by co-producing a social prescription—a non-medical prescription, to improve health and well-being and to strengthen community connections.'(p. 9).[11] This internationally accepted definition clearly outlines the diverse structural components of social prescribing and signifies the complex and multidimensional nature of the intervention.

Existing research does not describe what social prescribing interventions are currently in place for PLWD and/or their carers, and while quality of life has been shown to improve and extend through creative and social engagement, the mechanisms and process by which this can be achieved through social prescribing have not been explored.[12] Moreover, evidence on the needs that social prescribing is designed to address and the extent to which social prescribing interventions have been effective for PLWD and their carers is limited and heterogeneous making it difficult for conclusions to be drawn.[13] Little is known about the uptake of social prescribing interventions by PLWD and/or their carers, the reasons (circumstances) that lead them to participate in such interventions and the extent to which social prescribing interventions could potentially increase positive health outcomes if adopted and how.[13] To date, there is no systematic review of social prescribing for PLWD and/or

their carers (eg, family, friends, neighbours) to address these knowledge gaps.

This systematic review is designed to identify, describe and explore the mechanisms, processes and circumstances by which PLWD and/or their carers participate in social prescribing interventions. The research questions are:

1. What social prescribing interventions are currently available for PLWD and/or their carers in the UK?
2. Which PLWD and/or their carers are social prescribing interventions being delivered to?
3. What are the mechanisms (incl. services and agents) by which social prescribing interventions for PLWD and/or their carers are being instigated?
4. What are the processes through which PLWD and/or their carers receive social prescribing interventions?
5. For what reasons/circumstances do PLWD and/or their carers participate in social prescribing interventions?
6. What are the effects of social prescribing on (a) PLWD and/or their carers and (b) dementia-related healthcare and how are these measured?

This review has been designed as a complex intervention systematic review in accordance with a series of articles providing guidance and tools for reviews of complex interventions published by the Agency for Healthcare Research and Quality (AHRQ).[14–20] Focusing on uncovering and mapping the complexity of social prescribing in dementia care, logic modelling[15 16] will be employed to highlight multiple interacting components, causal relationships, non-linear pathways, variability in content, context and mode of delivery, as well as the unpredictability of impact and outcomes[21] of social prescribing interventions for PLWD and/or their carers.

## METHODOLOGY
### Paradigm
Logic modelling is a theory-based methodological paradigm, established and widely endorsed in programme evaluation research,[22] acting as a tool to illustrate and understand complexity.[23] Usually described as graphic representations of a system, or an intervention in this case, logic models, can be used to depict core intervention components, expected outcomes and the complex pathways through which these are connected. They can also be used to recognise the influence of contextual and implementation factors, explain causal relationships, describe any implicit or explicit theory of change[23] and identify gaps in existing knowledge.[24] Using a logic model is useful for planning, implementing and evaluating interventions, but also for making transparent assumptions and results more accessible to a wide range of key stakeholders.[25]

The role of logic models in evidence synthesis has been gaining attention, increasingly adopted in systematic reviewing, contributing both conceptually and analytically to most stages of the review process. Their ability to

conceptualise, map and explain complexity when systematically gathering, integrating and interpreting evidence from studies of complex interventions[22] makes logic modelling the most appropriate paradigm to use in this complex intervention systematic review.

## Approach

Considering the value that logic models can add to the different stages of the systematic review process,[26] three main approaches to logic modelling can be identified, that is, a priori, staged and iterative.[26] In the former, the logic model is constructed at protocol stage and finalised before any data is collected, not undergoing any further changes during the review process.[22] In the staged and iterative approach, the logic model is continuously revised, modified and adapted during the data collection process.[25 26] The decision of which approach to use is informed by the scope, aims and objectives of the systematic review.[26]

In this complex intervention systematic review, an iterative approach to logic modelling will be used whereby the initial logic model will go through various modifications as new data and knowledge is gained through the review process. The initial logic model will act as a guide to give 'an idea' of the key interacting components of the intervention.

Adaptation and modification of the initial logic model will continue throughout the review process as new insights emerge, which may identify new components, inter-relationships between components, move existing components to more appropriate positions or add scope to existing components. Finally, for the purposes of this complex intervention systematic review, the logic model will be revisited at the point of data extraction, data analysis and prior to dissemination, with clearly labelled versions of the logic model detailing how, and based on what information, changes have been made.

## Type

Within each of the three approaches to logic modelling, two types of models have been identified: system-based and process-oriented logic models. System-based logic models represent the system in which interactions occur but are predominantly static in the sense that, while interactions are identified, they are not explored in further detail.[25] System-based models focus on depicting the complexity of the components of the intervention rather than their interaction. In contrast, process-oriented logic models graphically represent the causal pathways leading from intervention to outcomes[25] concentrating more on the key relationships embodied and their interaction.[15]

Considering the aims of this review are to identify, illustrate, understand and explain the complexity of social prescribing as an intervention for PLWD and/or their carers, a process-orientated model will be used.

## Development of initial logic model

The initial process-oriented logic model will act as a guide for this complex intervention systematic review and was developed in the following six steps:

1. A model template including the target domains of the review (participants, intervention, outcomes, context and implementation) was identified as a starting point to facilitate the development of an initial logic model.[25]
2. This was shared with the team of multidisciplinary experts involved in the review to initiate brainstorming and conceptualisation. Through discussion and consensus, the target domains of the model were populated while literature searches revealed several relevant published frameworks, aspects of which were used to further populate some of the domains. In terms of context and implementation, relevant aspects were derived from the Context and Implementation of Complex Interventions Framework, a framework that consists of eight domains pertaining to context (ie, setting, geographical, epidemiological, sociocultural, socioeconomic, ethical, legal and political) and four domains pertaining to implementation (ie, provider, organisation and structure, funding and policy).[27] Additionally, working definitions for all target domains were created by the review team and mapped onto the initial logic model for clarity.
3. An initial system-based logic model was designed to represent the complexities of the different components of the intervention under review. To turn this into a process-oriented logic model, representing how the intervention could produce change, arrows were used to indicate causal relationships (pathways) between the different components and their interaction, drawing on the intervention Complexity Assessment Tool for Systematic Reviews.[28] This is a tool designed to assist reviewers in categorising, visually mapping and better understanding intervention complexity using a typology that will be then used to support data extraction, analysis and interpretation.
4. This logic model was considered by the SPLENDID Patient and Public Involvement (PPI) Advisory Group, to ensure that different perspectives were represented, and then refined based on the suggestions received.
5. The revised model was presented to the SPLENDID Work Package 1 Expert Advisory Group (researchers from a wide range of backgrounds including medicine, nursing, psychology and sociology) for further feedback.
6. The final version of the initial logic model was approved by the review team (figure 1).

## Application

The initial logic model, and its subsequent iterations, will be used as a model for data extraction, data analysis and dissemination.

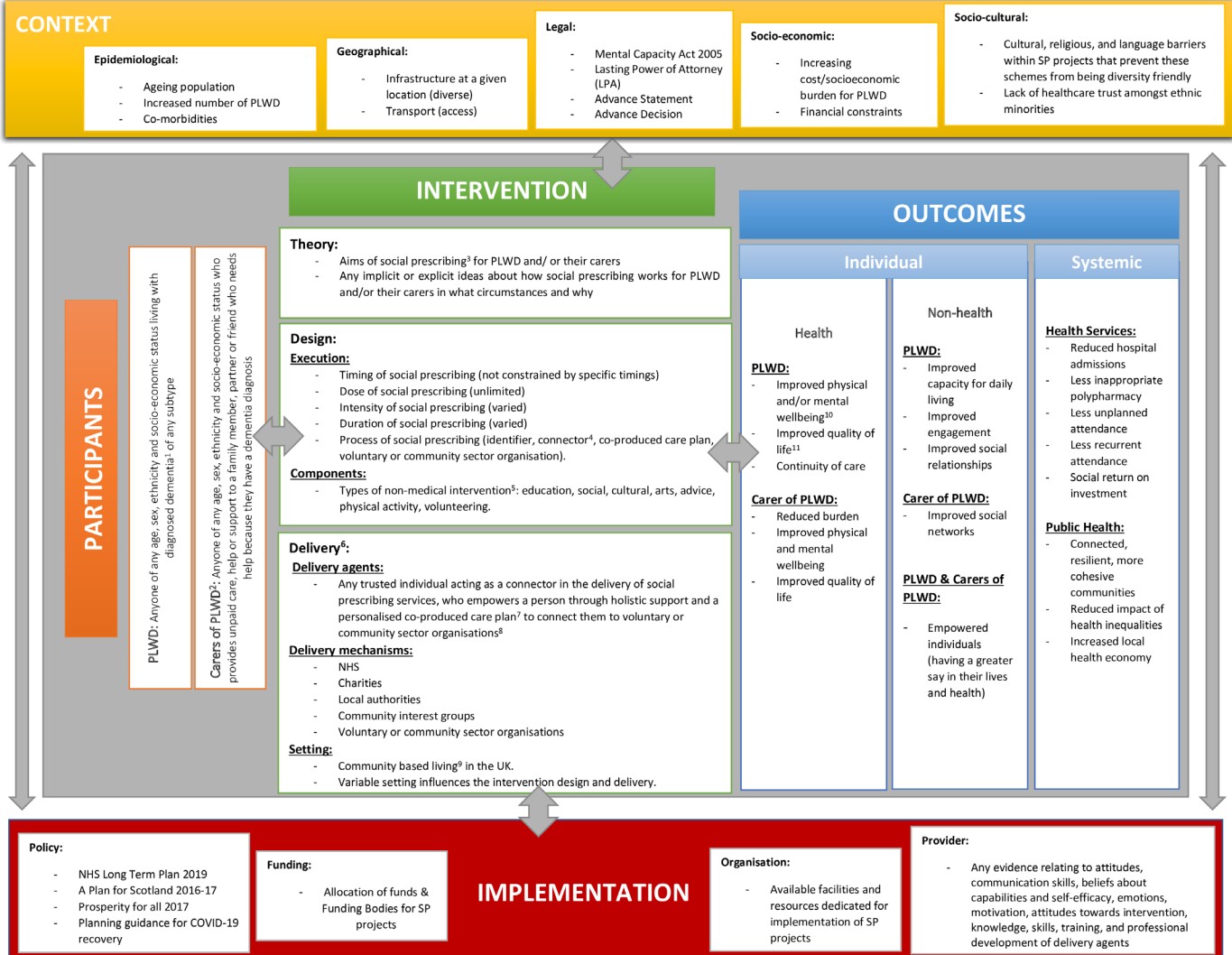

**Figure 1** Initial logic model and complex terminology definitions.

## METHODS

This protocol has been produced in line with the AHRQ series on complex intervention systematic review papers.[14–20] This review was registered with the International Register of Systematic Reviews (PROSPERO) on 16 June 2023 and will be reported following the Preferred Reporting Items for Systematic Review and Meta-Analysis Complex Interventions extension statement and checklist.[14–20]

Any subsequent modifications to the protocol will be recorded on PROSPERO and published in the findings of the review.

### Eligibility criteria

The PICOTS (Population, Intervention, Context, Outcomes, Timing and Setting) framework was used to define and develop a list of inclusion and exclusion criteria (table 1). PICOTS is an adaptation of the traditional PICO framework developed to map potential sources of complexity that can arise in a complex intervention systematic review, to which the addition of Timing

and Setting captures dimensions of complexity relating to contextual factors.[15]

### Information sources

#### Electronic searches

Comprehensive systematic literature searches were performed in June 2023 in collaboration with an information specialist on the following databases: MEDLINE (Ovid), EMBASE (Ovid), PsycINFO (EBSCO), CINAHL (EBSCO), Scopus (Elsevier) and Cochrane/CENTRAL (Cochrane Library), from 1 January 2003 to June 2023; hand searching of reference lists continued throughout 2023. This timeframe was selected to account for all research conducted in the past 20 years, considering that, while social prescribing emerged as a concept in the late 1990s, the practice gained prominence in the early 2000s through pilot programmes and initiatives in various parts of the UK. Grey literature will be searched via EThOS and CORE databases. Database searches will be supplemented by hand searching of relevant grey literature including web archives of charities and government websites, as

**Table 1** PICOTS table

| PICOTS | Inclusion | Exclusion |
|---|---|---|
| Population | *Person living with dementia:*<br>▲ Anyone of any age, sex, ethnicity and socioeconomic status living with diagnosed dementia (of any subtype or stage).<br>▲ Living in the community (residential care homes included), either independently or supported by others.<br>*Carers:*<br>▲ Of any age, sex, ethnicity and socioeconomic status.<br>▲ Providing unpaid care help or support to a family member, partner or friend living with diagnosed dementia (of any subtype or stage) but can be receiving a carer's allowance. | *Person living with dementia:*<br>▲ Not having a formal dementia diagnosis.<br>▲ Living in a nursing home, a hospice or receiving inpatient or end of life care.<br>*Carers:*<br>▲ Anyone providing care to PLWD not having a formal dementia diagnosis.<br>▲ Anyone providing paid care, help or support to PLWD (of any subtype or stage). |
| Intervention | Social prescribing (defined as): 'a means for trusted individuals in clinical and community settings to identify that a person has non-medical, health-related social needs and to subsequently connect them to non-clinical supports and services within the community by co-producing a social prescription – a non-medical prescription, to improve health and well-being and to strengthen community connections.(p. 9)'[11]<br>*Any non-medical interventions* (activities, groups or services within the community that meet the practical, social and emotional needs that affect health and well-being of patients living with dementia and/or their carers).<br>run by<br>*Any voluntary or community sector organisations* (either community-based or virtual) accessed through<br>*Any referral route* (such as primary care (GPs), charity, community interest group, local authority, self-referral, carer-referral but not limited to).<br>via a<br>*Connector* (a trusted individual in a clinical or community setting involved in the delivery of social prescribing services)<br>through a<br>*Personalised co-produced care plan* (a verbal or written holistic, person-centred adaptive plan to address non-medical health-related needs based on 'what matters' to a person) | ▲ Any medical interventions (eg, pharmacological).<br>▲ Any non-medical interventions accessed without a connector being involved and no personalised co-produced care plan used. |
| Context | *Published output:*<br>▲ Focusing on UK healthcare.<br>▲ Papers of any empirical design.<br>▲ Grey literature (reports).<br>▲ January 2003–October 2023. | *Published output:*<br>▲ Focusing on International healthcare.<br>▲ Reviews.<br>▲ Case studies.<br>▲ Editorials.<br>▲ News.<br>▲ Conference abstracts and proceedings. |
| Outcomes | Any quantitative (eg, scores of psychometric instruments and/or other standardised measures) or qualitative (perceptions, thoughts, experiences) individual (health and non-health) and systemic outcomes, including proxy perspectives (see figure 1). | No restriction |
| Timing | No time restriction | No time restriction |
| Setting | Community-based living anywhere in the UK. | PLWD living in a nursing home, a hospice or receiving inpatient or end of life care. |

PLWD, people living with dementia.

well as reference lists and citations of included studies. The search strategy will include text words and database-specific subject headings relating to each of the following areas: 'dementia', 'social prescribing', 'assessment', 'referral', 'link worker' and the 'UK' combined and tailored to the relevant database. The search strategy will be limited to studies conducted in the UK, due to the variation in coverage and delivery of social prescribing. Only studies reported in the English language will be included.

## Data collection and analysis
### Study selection
Once the searches have been completed all studies identified will be extracted and imported to EndNote V.20 for deduplication. Unique citations will be imported to Rayyan,[29] the Intelligent Systematic Review Software, for screening. Title and abstracts will be independently screened against inclusion/exclusion criteria by two members of the review team. Full texts of potentially eligible papers identified during title and abstract screening will be independently assessed by at least two reviewers. Any discrepancies will be resolved through consensus-based decisions or, if necessary, in consultation with a third reviewer.

### Assessment of risk of bias
Gough's Weight of Evidence,[30] a framework for the appraisal of the quality and relevance of evidence, applicable to both quantitative and qualitative papers, will be used to identify any risk of bias in the included studies by assessing study quality and relevance. Any disagreements of scores in paper reviewing and assessment will be resolved through consensus-based decisions and in consultation with a third reviewer, where necessary.

### Data extraction and management
The initial process-based logic model will be used to guide data extraction. A visual map of extracted information (using an Excel spreadsheet), coded under different domains of the model, based on the research questions addressed, will be constructed, and used to explore and account for the mechanisms, processes and circumstances by which PLWD and/or their carers participate in social prescribing interventions.

Information to be extracted will include:
1. Participants (ie, PLWD, family/carers and providers) involved in the intervention.
2. Mechanisms of intervention delivery.
3. Implementation data (processes of intervention delivery).
4. Multilevel contextual factors that influence implementation, intervention mechanisms and outcome.
5. Outcome measures from the intervention.

To ensure consistency and minimise bias of data extraction, 10% of included papers will be independently assessed by at least two members of the review team. The initial logic model will be reviewed at this stage and updated as necessary. Following that, data extraction for all included papers will continue to be carried out by at least two reviewers. The review team will meet regularly to discuss and resolve any differences in data extraction through discussion and consensus.

### Data synthesis
Due to the complexity of the topic addressed, heterogeneity of evidence identified is anticipated. Findings will, therefore, be integrated using narrative synthesis.[31] This approach was selected for its potential to synthesise heterogeneous and complex evidence in a rigorous and replicable way. Findings will be conceptually triangulated using the initial process-based logic model as a guide to construct groupings of evidence that are conceptually similar and identify and explore the relationships between these groupings. The SPLENDID PPI and SPLENDID Work Package 1 Expert Advisory Group will regularly offer guidance on data synthesis, mapping and interpretation to ensure pertinent issues are identified and addressed.

### Analysis of subgroups or subsets
Data permitting, subgroup analysis will be performed. It is not possible to specify the groups in advance.

## SPLENDID PPI Group and SPLENDID Work Package 1 Expert Advisory Group
### Patient and public involvement statement
The SPLENDID PPI and the SPLENDID Work Package 1 Expert Advisory Groups will be consulted at various stages of the review. Thus far, the groups have contributed to refining the review questions and the initial logic model by reviewing their relevance in the context of their own personal and professional experience. Any pertinent issues identified relating to contextual factors, implementation challenges and outcomes, for which revisions were required, were considered and revised accordingly. The groups will reconvene at the data synthesis stage to offer guidance to assess the preliminary themes identified and review their relevance. A final meeting will take place towards the end of the review for the groups to review and assess the reporting of findings and provide guidance on dissemination plans.

## ETHICS AND DISSEMINATION
### Ethical considerations
This is a systematic review that strictly operates with secondary sources of data openly accessible in the public domain; therefore, no ethical approval or consent will be required.

### Dissemination plan
To achieve widespread dissemination of the findings of this review, the following activities have been put in place:
1. Publishing in open access, peer-reviewed journals.
2. Presenting at relevant national and international conferences and other events.

3. Organising dissemination meetings with key stakeholders such as healthcare professionals, patient and carer groups, community organisations (eg, the Social Prescribing Network and the Evidence Collaborative at the National Academy for Social Prescribing), policymakers and funding bodies.

## GLOSSARY OF INITIAL LOGIC MODEL TERMS & DEFINTIONS*

1. Dementia: a diagnosed syndrome related to cognitive and behavioural decline which over time affects memory, problem solving, language, mood, motivation and behaviour to such an extent that it interferes with a person's daily life and activities.

2. Carers of PLWD: anyone of any age who provides unpaid care, help or support to a family member, partner or friend who needs help because they have a dementia diagnosis.

3. Social prescribing: a means for trusted individuals in clinical and community settings to identify that a person has non-medical, health-related social needs and to subsequently connect them to non-clinical supports and services within the community by co-producing a social prescription—a non-medical prescription, to improve health and well-being and to strengthen community connections (p. 9).[11]

4. Connector: a trusted individual in a clinical or community setting involved in the delivery of social prescribing services, who empowers a person through holistic support and a personalised co-produced care plan to connect them to voluntary and community sector organisations.

5. Non-medical intervention: a non-clinical approach, activity, opportunity or support scheme.

6. Delivery: intervention delivery describes the 'how' (delivery mechanisms), 'who' (delivery agents), 'where' (setting) of the intervention.[25]

7. Personalised co-produced care plan: a verbal or written holistic, person-centred adaptive plan based on 'what matters' to a person. It is produced in equal partnership with a trusted individual in a clinical or community setting to address non-medical health-related needs.

8. Voluntary and community sector organisations: a not for profit, non-clinical community asset or scheme that offers people in-person or remote services, activities, opportunities or support.

9. Community-based living: living primarily in the community, either independently or supported by others, including in a residential care home, but not in a nursing home, a hospice or receiving inpatient or end of life care.

10. Well-being: exists in two dimensions. Subjective well-being relates to how people feel and how they function on a personal level in relations to their financial, health, social, personal and local environment. Objective well-being relates to objective measures of an individual's being and assumptions about basic human needs and rights.

11. Quality of life: a multidimensional measure of an individual's health in terms of (but not limited to) their physical, psychological, social, personal and environmental state.

*Working definitions of terms for which inconsistency and/or ambiguity persists in the current literature were specifically developed drawing on stakeholder conversations, clinical, peer reviewed and grey literature, being adapted to fit the aims of this Complex Intervention Systematic Review (CISR).

**Author affiliations**
[1]Faculty of Medicine and Health Sciences, University of East Anglia, Norwich, UK
[2]University of Leeds, Leeds, UK
[3]University of Exeter, Exeter, UK
[4]University of East Anglia, Norwich, UK
[5]University of Hull, Hull, UK
[6]University of Newcastle upon Tyne, Newcastle upon Tyne, UK
[7]Meaningful Measures Ltd, Bristol, UK
[8]Hull York Medical School, Hull, UK
[9]Institute for Health and Society, Newcastle University, Newcastle, UK
[10]Faculty of Environmental and Life Sciences, School of Health Sciences, University of Southampton, Southampton, UK
[11]Faculty of Health Sciences, University of Hull, Hull, UK

**Correction notice** This article has been corrected since it was published. The missing Acknowledgements section has been added and the author list has been updated.

**Acknowledgements** The authors would like to give special thanks to the SPLENDID Patient and Public Involvement Advisory Group, the SPLENDID Work Package 1 Expert Advisory Group, and the members of the SPLENDID Collaboration for their support and contribution to this work. We would also like to thank our charity partners TIDE and DEEP, and the Norwich Clinical Trials Unit, for their support. The contract for this research is held by the Norfolk and Waveney Integrated Care Board (NWICB) and it is supported by the National Institute for Health and Care Research (NIHR) Applied Research Collaborations of the East of England, Wessex, and South West Peninsula. The views expressed in this publication are those of the authors and not necessarily those of the NWICB, the National Institute for Health and Care Research or the Department of Health and Social Care.

**Collaborators** The SPLENDID Collaboration: Prof Chris Fox, Dr Jane Cross, Prof Louise Allan, Prof Anthony Avery, Dr Katherine Bradbury, Anne Irvine, Jessica Marshall, Prof Antonieta Medina-Lara, Prof Esme Moniz-Cook, Nia Morrish, Prof Martin Orrell, Dr Evie Papavasiliou, Aimee Pick, Prof Fiona Poland, Dr Marie Polley, Dr Amy Rathbone, Prof Joanne Reeve, Prof Dame Louise Robinson, George Rook, Dr Euan Sadler, Dr Kritika Samsi, Prof Lee Shepstone, Dr Sarah Walker, Dr Emma Wolverson.

**Contributors** JLC and CF contributed to funding acquisition. JM, EP, SW and JLC contributed to conceptualisation. JM, EP, SW and JLC contributed to methodology. JM, EP, CF, AI, EM-C, AP, MJP, JR, LR, GR, ES, EW, SW and JLC contributed to the development of the logic model. SPLENDID PPI Group: AI and GR. SPLENDID Work Package 1 Expert Advisory Group: CF, EM-C, MJP, JR and ES. JM, MH and EP contributed to the development of search strategy. JM, EP, SW and JLC contributed to writing–original draft preparation. JM, EP, CF, MH, AI, EM-C, AP, MJP, JR, LR, GR, ES, EW, SW and JLC have reviewed and edited the protocol. JM, EP, CF, MH, AI, EM-C, AP, MJP, JR, LR, GR, ES, EW, SW and JLC have approved submission(s).

**Funding** This study/project is funded by the NIHR Programme Grants for Applied Research [NIHR203280].

**Disclaimer** The views expressed are those of the author(s) and not necessarily those of the NIHR or the Department of Health and Social Care.

**Competing interests** Minor competing interest declared by co-author (EMC) regarding salary costs from the sponsor funded by the NIHR.

**Patient and public involvement** Patients and/or the public were involved in the design, or conduct, or reporting, or dissemination plans of this research. Refer to the Methods section for further details.

**Patient consent for publication** Not applicable.

**Provenance and peer review** Not commissioned; externally peer reviewed.

**ORCID iDs**
Jessica Marshall http://orcid.org/0009-0001-8234-6077
Evie Papavasiliou http://orcid.org/0000-0002-6504-515X
Chris Fox http://orcid.org/0000-0001-9480-5704
Matthew Hawkes http://orcid.org/0000-0001-7566-4555
Esme Moniz-Cook http://orcid.org/0000-0002-7232-4632
Aimee Pick http://orcid.org/0009-0008-6361-2297
Marie J Polley http://orcid.org/0000-0003-3823-5668
Joanne Reeve http://orcid.org/0000-0002-3144-7955
Louise Robinson http://orcid.org/0000-0003-0209-2503
Euan Sadler http://orcid.org/0000-0003-3827-224X
Emma Wolverson http://orcid.org/0000-0002-5789-7862
Sarah Walker http://orcid.org/0000-0003-4201-1093
Jane L Cross http://orcid.org/0000-0002-7003-1916

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
