## [Reviewer comments · BMJ Open]

ARTICLE DETAILS

TITLE (PROVISIONAL)	Social prescribing for People Living with Dementia (PLWD) and their carers: what works, for whom, under what circumstances, and why: Protocol for a complex intervention systematic review
AUTHORS	Marshall, Jessica; Papavasiliou, Evie; Fox, Chris; Hawkes, Matthew; Irvine, Anne; Moniz-Cook, Esme; Pick, Aimee; Polley, Marie; Reeve, Joanne; Robinson, Louise; Rook, George; Sadler, EA; Wolverson, Emma; Walker, Sarah; Cross, Jane

VERSION 1 – REVIEW

REVIEWER	Muhl, Caitlin Queen's University
REVIEW RETURNED	16-Nov-2023

GENERAL COMMENTS	This protocol is well written. This complex intervention systematic review will make a valuable contribution to the social prescribing literature. I have a few comments: 1. Given that social prescribing is taking place in 30+ countries across the globe, it would be helpful to add 1-2 sentences to justify why you chose to limit your review to the UK.2. I suggest using another word instead of 'exploit'.3. I noticed that some references are missing the journal title and/or DOI so the reference list needs to be checked over to make sure that everything has been formatted properly. I look forward to seeing this work published.
--

REVIEWER	Guion , Vincent Centre Hospitalier Universitaire de Toulouse Gerontopole
REVIEW RETURNED	27-Nov-2023

GENERAL COMMENTS	Thank you for giving me the opportunity to learn more on social prescribing and on complex evaluation methods. I do not feel qualified to revise the most complex details of the method, but the rest of the manuscript displays good reasons to trust authors in their choices. I found a couple of typos: - Page 7/20, L44: acts acted (choose one)- Figure 1: The PLWD orange box cut the bottom half of the last phrase.
---

REVIEWER	Swaffer, Kate Dementia Alliance International
-----------------	--

REVIEW RETURNED	29-Nov-2023
GENERAL COMMENTS	VERY minor edits would improve this paper
REVIEWER	Eldor, Roy Tel Aviv Sourasky Medical Center, Diabetes Unit
REVIEW RETURNED	15-Jan-2024
GENERAL COMMENTS	Marshall et. al describe a protocol for a systematic review of studies addressing social prescribing in people living with dementia. The topic and the suggested review of the literature are important and will bridge a key knowledge gap. The methodology and methods are sound. Comments 1. Abstract The abstract does not clearly state that the systematic review will focus on SP in PLWD. Moreover, it does not mention the search criteria but only the search methods. 2. Typo in line 44 on page 7.

VERSION 1 – AUTHOR RESPONSE

Reviewer 1 Comment (CM)	Given that social prescribing is taking place in 30+ countries across the globe, it would be helpful to add 1-2 sentences to justify why you chose to limit your review to the UK.	The search strategy will be limited to English language works only. (page 8 line 45-46)	Actioned: justification added (social prescribing coverage and delivery are highly variable).	Covered in point above.
Reviewer 1 Comment (CM)	I suggest using another word instead of 'exploit'.	As part of the dissemination plan to exploit the findings produced by this review various activities have been put in place including: (page 10 line 51-52)	Actioned: this phrase has been updated to not include the word 'exploit'.	To achieve widespread dissemination of the findings of this review, the following activities have been put in place:
Reviewer 1 Comment (CM)	I noticed that some references are missing the journal title and/or DOI so the reference list needs to be checked over	Muhl C, Mulligan K, Bayoumi I, et al. Establishing Internationally Accepted Conceptual and Operational Definitions of Social Prescribing Through Expert Consensus: A Delphi Study.	Actioned: Journal title added to Muhl et al reference. Actioned: DOIs added to Tierney et al,	Muhl C, Mulligan K, Bayoumi I, et al. Establishing Internationally Accepted Conceptual and Operational Definitions of Social Prescribing Through Expert Consensus: A Delphi Study. BMJ Open. 2023.

	to make sure that everything has been formatted properly.	2023. 13:e070184. doi: 10.1136/bmjopen-2022-070184 (page 13 line 9-12) Tierney S, Wong G, Roberts N, , et al. Supporting social prescribing in primary care by linking people to local assets: a realist review. BMC Medicine. 2020; 13;18(1) (page 13 line 6-7) Pfadenhauer LM, Gerhardus A, Mozygemba K, et al. Making sense of complexity in context and implementation: the Context and Implementation of Complex Interventions (CICI) framework. Implementation Science. 2017; 15;12(1). (page 14 line 18-21) Lewin S, Hendry M, Chandler J, , et al. Assessing the complexity of interventions within systematic reviews: development, content and use of a new tool (iCAT_SR). BMC Medical Research Methodology. 2017;26;17(1). (page 14 line 23-26) Ouzzani M, Hammady H, Fedorowicz Z, et al. Rayyan—a web and mobile app for systematic reviews. Systematic Reviews. 2016;5(1). https://doi-org.uea.idm.oclc.org/10.1186/s13643-016-0384-4 (page 14 line 27-29)	Pfadenhauer et al, Lewin et al and Ouzzani et al. Actioned: Additional minor typos and grammar changes made throughout the document (tracked changes will reflect these).	13:e070184. doi: 10.1136/bmjopen-2022-070184 Tierney S, Wong G, Roberts N, , et al. Supporting social prescribing in primary care by linking people to local assets: a realist review. BMC Medicine. 2020; 13;18(1). doi: 10.1186/s12916-020-1510-7. Pfadenhauer LM, Gerhardus A, Mozygemba K, et al. Making sense of complexity in context and implementation: the Context and Implementation of Complex Interventions (CICI) framework. Implementation Science. 2017; 15;12(1). doi: 10.1186/s13012-017-0552-5 Lewin S, Hendry M, Chandler J, , et al. Assessing the complexity of interventions within systematic reviews: development, content and use of a new tool (iCAT_SR). BMC Medical Research Methodology. 2017;26;17(1). doi: 10.1186/s12874-017-0349-x Ouzzani M, Hammady H, Fedorowicz Z, et al. Rayyan—a web and mobile app for systematic reviews. Systematic Reviews. 2016;5(1). doi: 10.1186/s13643-016-0384-4
Reviewer 2 Comment (VG)	Page 7/20, L44: acts acted (choose one)	The initial process-oriented logic model acts acted as the guide for this complex intervention systematic review and was developed in the following six steps (page 7 line 44)	Actioned: Line re-phrased.	The initial process-oriented logic model will act as a guide for this complex intervention systematic review and was developed in the following six steps:

Reviewer 2 Comment (VG)	Figure 1: The PLWD orange box cut the bottom half of the last phrase.		Actioned: PLWD orange box reformatted so no phrases are cut off and re-submitted as a supplementary file.	See revised supplementary information.
Reviewer 4 Comment (RE)	Abstract: The abstract does not clearly state that the systematic review will focus on SP in PLWD. Moreover, it does not mention the search criteria but only the search methods.	A complex intervention systematic review will be conducted using an iterative logic model approach (page 2 line 29-30)	Actioned: stated clearer the review's focus. However, considering word limit and the complex nature of the review we cannot explain the search criteria in the abstract.	A complex intervention systematic review of SP for PLWD and/or their carers will be conducted using an iterative logic model approach.
Reviewer 4 Comment (RE)	Typo in line 44 on page 7.	The initial process-oriented logic model acts as the guide for this complex intervention systematic review and was developed in the following six steps (page 7 line 44)	Actioned: Line re-phrased.	Addressed above in point 9
Author Revision	Add new authors and ORCID IDs	Authors: Jessica Marshall, University of East Anglia, j.marshall@uea.ac.uk Dr Evie Papavasiliou, University of Leeds, e.papavasiliou@leeds.ac.uk Prof. Chris Fox, University of Exeter, christopher.fox@exeter.ac.uk Prof. Esme Moniz-Cook, University of Hull, e.d.moniz-cook@hull.ac.uk Dr Marie Polley, Meaningful Measures Ltd, marie@meaningfulmeasures.co.uk Prof. Joanne Reeve, Hull York Medical School, joanne.reeve@hyms.ac.uk Dr Euan Sadler, University of Southampton, e.a.sadler@soton.ac.uk	Actioned: authors that were missed in the previous submission have been added to the manuscript and will be added on BMJ Open when the manuscript is resubmitted. Co-author ORCID IDs also added to the manuscript.	New Authors List: Jessica Marshall, University of East Anglia, j.marshall@uea.ac.uk ORCID ID: 0009-0001-8234-6077 Dr Evie Papavasiliou, University of Leeds, e.papavasiliou@leeds.ac.uk ORCID ID: 0000-0002-6504-515X Prof. Chris Fox, University of Exeter, christopher.fox@exeter.ac.uk ORCID ID: 0000-0001-9480-5704 Matthew Hawkes, University of East Anglia, m.hawkes@uea.ac.uk Anne Irvine, University of East Anglia, anne.irvine10@outlook.com

		Matthew Hawkes, University of East Anglia, m.hawkes@uea.ac.uk Dr Sarah Walker, University of Exeter, s.walker@exeter.ac.uk Dr Jane Cross, University of East Anglia, j.cross@uea.ac.uk (page 1 line 15-35)		Prof. Esme Moniz-Cook, University of Hull, e.d.moniz- cook@hull.ac.uk ORCID ID: 0000-0002- 7232-4632 Aimee Pick, Newcastle University, aimee.pick@newcastle.a c.uk ORCID ID: 0009-0008- 6361-2297 Dr Marie Polley, Meaningful Measures Ltd, marie@meaningfulmeas ures.co.uk ORCID ID: 0000-0003- 3823-5668 Prof. Joanne Reeve, Hull York Medical School, joanne.reeve@hyms.ac. uk ORCID ID: 0000-0002- 3184-7955 Prof. Dame Louise Robinson, Newcastle University, a.l.robinson@newcastle. ac.uk ORCID ID: 0000-0003- 0209-2503 George Rook, University of East Anglia, georgerook51@gmail.co m Dr Euan Sadler, University of Southampton, e.a.sadler@soton.ac.uk ORCID: 0000-0003- 3827-224X Dr Emma Wolverson, University of Hull, e.wolverson@hull.ac.uk ORCID ID: 0000-0002- 5789-7862 Dr Sarah Walker, University of Exeter, s.walker@exeter.ac.uk ORCID ID: 0000-0003- 4201-1093 Dr Jane Cross, University of East Anglia, j.cross@uea.ac.uk ORCID ID: 0000-0002- 7003-1916
Author Revision	Add the names of the		Actioned: SPLENDID Collaboratio	On behalf of the SPLENDID Collaboration

	SPLENDID Collaboration		n names added.	Prof. Chris Fox, Dr Jane Cross, Prof. Louise Allan, Prof. Anthony Avery, Dr Katherine Bradbury, Anne Irvine, Jessica Marshall, Prof. Antonieta Medina-Lara, Prof. Esme Moniz-Cook, Nia Morrish, Prof. Martin Orrell, Dr Evie Papavasiliou, Aimee Pick, Prof. Fiona Poland, Dr Marie Polley, Dr Amy Rathbone, Prof. Joanne Reeve, Prof. Dame Louise Robinson, George Rook, Dr Euan Sadler, Dr Kritika Samsi, Prof. Lee Shepstone, Dr Sarah Walker, Dr Emma Wolverson
Author Revision	Add funding statement according to SPLENDID Publication Strategy & Policy (see emails)	This study/project is funded by the NIHR Programme Grants for Applied Research [NIHR203280]. The views expressed are those of the author(s) and not necessarily those of the NIHR or the Department of Health and Social Care. (page 11 line 26-32)	Actioned: Replace old funding statement with new funding statement.	This study/project is funded by the NIHR Programme Grants for Applied Research [NIHR203280]. The views expressed are those of the author(s) and not necessarily those of the NIHR or the Department of Health and Social Care.
Author Revision	Amend 'Expert Advisory Group' to 'SPLENDID Work Package 1 Expert Advisory Group'.	The revised model was presented to the Expert Advisory Group of the SPLENDID project (researchers from a wide range of backgrounds including medicine, nursing, psychology, and sociology) (page 7 line 32) The PPI and Expert Advisory Group will regularly offer guidance on data synthesis (page 10 line 8) Patient and Public Involvement (PPI) & Expert Advisory Group (page 10 line 21) Patient and Public Involvement (PPI) and Expert Advisory Groups will be consulted at various stages of the review (page 10 line 23)	Actioned: Instances where the 'Expert Advisory Group' are referenced in the manuscript have been replaced by the 'SPLENDID Work Package 1 Expert Advisory Group'.	The revised model was presented to the SPLENDID Work Package 1 Expert Advisory Group (researchers from a wide range of backgrounds including medicine, nursing, psychology, and sociology) The PPI and SPLENDID Work Package 1 Expert Advisory Group Patient and Public Involvement (PPI) & SPLENDID Work Package 1 Expert Advisory Group Patient and Public Involvement (PPI) and the SPLENDID Work Package 1 Expert Advisory Groups will be consulted at various stages of the review

			The authors would like to give special thanks to the Patient and Public Involvement Advisory Group and the Expert Advisory Group for their support and contribution to this work. (page 11 line 15)		The authors would like to give special thanks to the Patient and Public Involvement Advisory Group and the SPLENDID Work Package 1 Expert Advisory Group for their support and contribution to this work.
Author Revision	Amend 'PPI' to 'SPLENDID PPI Group'	This logic model was considered by a Patient and Public Involvement (PPI) Advisory Group, to ensure that different perspectives were represented, and then refined based on the suggestions received. (page 7 line 27-30) The PPI and Expert Advisory Group will regularly offer guidance on data synthesis (page 10 line 8) Patient and Public Involvement (PPI) & Expert Advisory Group (page 10 line 21) Patient and Public Involvement (PPI) and Expert Advisory Groups will be consulted at various stages of the review (page 10 line 23) PPI and expert advice: AI, GR, CF, JR, MP, ES, EMC (page 11 line 19) The authors would like to give special thanks to the Patient and Public Involvement Advisory Group and the SPLENDID Work Package 1 Expert Advisory Group for their support and contribution to this work. (page 11 line 13)	Actioned: Instances when 'PPI' was mentioned in the manuscript have been replaced by 'SPLENDID PPI Group'.	This logic model was considered by the SPLENDID Patient and Public Involvement (PPI) Advisory Group, to ensure that different perspectives were represented, and then refined based on the suggestions received. The SPLENDID PPI Group and Expert Advisory Group will regularly offer guidance on data synthesis SPLENDID Patient and Public Involvement (PPI) Group & Expert Advisory Group (page 10 line 21) The SPLENDID Patient and Public Involvement (PPI) and Expert Advisory Groups will be consulted at various stages of the review SPLENDID PPI Group: AI, GR. The authors would like to give special thanks to the SPLENDID Patient and Public Involvement Advisory Group and the SPLENDID Work Package 1 Expert Advisory Group for their support and contribution to this work.	
Author Revision	Add new authors' contributions paragraph to reflect	AUTHORS' CONTRIBUTIONS. Funding acquisition: JC and CF; Conceptualisation: JM, EP, SW and JC;	Actioned: Authors' Contributions paragraph reformatted	AUTHORS' CONTRIBUTIONS. Funding acquisition: JC and CF; Conceptualisation: JM,	

	contributions from added authors	Methodology: JM, EP, SW and JC; Development of logic model: JM, EP and SW. PPI and expert advice: AI, GR, CF, JR, MP, ES, EMC.; Development of search strategy: MH and JM.; Writing – original draft preparation: JM, EP, SW, JC. JM, EP, CF, EMC, MP, ES, JR, SW, JC have reviewed and edited the protocol; JM, EP, CF, EMC, MP, ES, JR, MH, SW, JC have approved submission. (page 11 line 17-24)	with added contributions by various authors.	EP, SW and JC; Methodology: JM, EP, SW JC; Development of the logic model: JM, EP, CF, AI, EMC, AP, MP, JR, LR, GR, ES, EW, SW, JC; SPLENDID PPI Group: AI, GR; SPLENDID Work Package 1 Expert Advisory Group: CF, EMC, MP, JR, ES; Development of search strategy: JM, MH, EP; Writing – original draft preparation: JM, EP, SW, JC; Have reviewed and edited the protocol; JM, EP, CF, MH, AI, EMC, AP, MP, JR, LR, GR, ES, EW, SW, JC; Have approved submission(s): JM, EP, CF, MH, AI, EMC, AP, MP, JR, LR, GR, ES, EW, SW, JC.
Author Revision	Amend the 'Acknowledgments' paragraph to reflect the requirements of the SPLENDID Publication Strategy.	The authors would like to give special thanks to the Patient and Public Involvement Advisory Group and the Expert Advisory Group for their support and contribution to this work. (page 11 line 13-15)	Actioned: The SPLENDID Collaboration was added in the Acknowledgment paragraph.	The authors would like to give special thanks to the SPLENDID Patient and Public Involvement Advisory Group, the SPLENDID Work Package 1 Expert Advisory Group, and the members of the SPLENDID Collaboration for their support and contribution to this work. We would also like to thank our charity partners TIDE and DEEP, and the Norwich Clinical Trials Unit, for their support. The contract for this research is held by the Norfolk and Waveney Integrated Care Board (NWICB) and it is supported by the National Institute for Health and Care Research (NIHR) Applied Research Collaborations of the East of England, Wessex, and South West Peninsula. The views expressed in this publication are those of the authors and not

					necessarily those of the NWICB, the National Institute for Health and Care Research or the Department of Health and Social Care.
Author Revision	Abstract reworded to stay in word count	Introduction: Dementia is a complex medical condition that poses significant challenges to healthcare systems and support services, affecting millions of people worldwide. People living with dementia (PLWD) and their carers experience multifaceted, complex needs often exacerbated by social isolation and challenges in accessing support. Social prescribing (SP) seeks to enable PLWD and their carers to access community and voluntary sector resources to support them address such needs. Existing research, however, does not describe what SP interventions are currently in place in dementia care. Little is also known about the needs these interventions are designed to address, the reasons that lead PLWD and their carers to participate in them, their effectiveness and the extent to which they could potentially increase positive health outcomes if adapted and how. Methods and Analysis: A complex intervention systematic review of social prescribing for people living with dementia and/or their carers will be conducted using an iterative logic model approach. Six electronic (MEDLINE, EMBASE, PsycINFO, CINAHL, Scopus, and Cochrane / CENTRAL) and two grey literature databases (EThOS and CORE) will be searched, supplemented by handsearching of reference lists of included studies.	Actioned: Reduced word count in the Abstract	Introduction: Dementia is a complex medical condition that poses significant challenges to healthcare systems and support services., affecting millions of people worldwide. People living with dementia (PLWD) and their carers experience multifaceted, complex needs often exacerbated by social isolation and challenges in accessing support. Social prescribing (SP) seeks to enable PLWD and their carers to access community and voluntary sector resources to support them address such needs. Existing research, however, does not describe what SP interventions are currently in place in dementia care. Little is also known about the needs these interventions are designed to address, the reasons that lead PLWD and their carers to participate in them, their effectiveness, and the extent to which they could potentially increase positive health outcomes if adapted adopted and how. Methods and Analysis: A complex intervention systematic review of SP for PLWD and/or their carers will be conducted using an iterative logic model approach. Six electronic (MEDLINE, EMBASE, PsycINFO, CINAHL, Scopus, and Cochrane / CENTRAL) and two grey literature databases (EThOS and	

		Study selection, data extraction, and risk of bias assessment, using Gough's Weight of Evidence Framework, will be independently performed by two reviewers. A narrative approach will be employed to synthesise and report quantitative and qualitative data. Reporting will be informed by the Preferred Reporting Items for Systematic Review and Meta-Analysis Complex Interventions (PRISMA-CI) extension statement and checklist. Ethics and Dissemination: No ethical approval is required due to this systematic review operating only with secondary sources. Findings will be disseminated through peer-reviewed publications, conference presentations and meetings with key stakeholders including healthcare professionals, patient and carer groups, community organisations (e.g., the Social Prescribing Network, and the Evidence Collaborative at the National Academy for Social Prescribing/NASP), policymakers, and funding bodies.		CORE) werewill be searched for publications between 1st January 2003 to June 2023, supplemented by handsearching of reference lists of included studies. Study selection, data extraction, and risk of bias assessment, using Gough's Weight of Evidence Framework, will be independently performed by two reviewers. A narrative approach will be employed to synthesise and report quantitative and qualitative data. Reporting will be informed by the Preferred Reporting Items for Systematic Review and Meta-Analysis Complex Interventions (PRISMA-CI) extension statement and checklist. Ethics and Dissemination: No ethical approval is required due to this systematic review operating only with secondary sources. Findings will be disseminated through peer-reviewed publications, conference presentations and meetings with key stakeholders including healthcare professionals, patient and carer groups, community organisations (e.g., the Social Prescribing Network, and the Evidence Collaborative at the National Academy for Social Prescribing/NASP), policymakers, and funding bodies.
BMJ Open Revision	Patient and Public Involvement Statement sub heading		Actioned: statement added	Patient and Public Involvement Statement

		added in Methods			
BMJ Open Revision	Please cite your figure as Figure 1 and not as appendix, figure should be re-uploaded in 1 page only, in PDF format. Please include figure legend at the end of your main manuscript, be informed that the figure legend or figure caption is a title or brief explanation appended to your figure.		Actioned: Removed appendix from the text and re-uploaded as PDF on submission. However, this figure needs to go across two pages, it cannot be reduced to one. Actioned: Figure legend added to the bottom of the manuscript.	(Figure 1.) Figure 1. Initial Logic Model and complex terminology definitions	
BMJ Open Revision	Kindly embed your table in your manuscript (should be editable and in table tools format). Table should be placed in the main text where the table is first cited. Please delete the separately uploaded table to avoid confusion. Please also cite your table as Table 1 only and not as appendix.			(Table 1) [Table added within text in CLEAN VERSION of manuscript]	
Author Revision	Update word count to account for revisions	Word Count: 3372 (page 1 line 39)		Word Count: 3147